# The Prevalence and Correlates of Disability in Singapore: Results from a Nationwide Cross-Sectional Survey

**DOI:** 10.3390/ijerph182413090

**Published:** 2021-12-11

**Authors:** Mythily Subramaniam, Yen Sin Koh, P. V. AshaRani, Fiona Devi, Saleha Shafie, Peizhi Wang, Edimansyah Abdin, Janhavi Ajit Vaingankar, Chee Fang Sum, Eng Sing Lee, Siow Ann Chong

**Affiliations:** 1Research Division, Institute of Mental Health, 10 Buangkok View, Singapore 539747, Singapore; Yen_Sin_KOH@imh.com.sg (Y.S.K.); Asharani_PEZHUMMOOTTIL_VASUDEVAN_N@imh.com.sg (P.V.A.); Fiona_Devi_Siva_Kumar@imh.com.sg (F.D.); saleha_shafie@imh.com.sg (S.S.); peizhi_wang@imh.com.sg (P.W.); Edimansyah_Abdin@imh.com.sg (E.A.); janhavi_vaingankar@imh.com.sg (J.A.V.); siow_ann_chong@imh.com.sg (S.A.C.); 2Saw Swee Hock Public Health, National University of Singapore, 12 Science Drive 2, Singapore 117549, Singapore; 3Admiralty Medical Centre, Khoo Teck Puat Hospital, 676 Woodlands Drive 71, Singapore 730676, Singapore; sum.chee.fang@ktph.com.sg; 4National Healthcare Group Polyclinics, 3 Fusionopolis Link, Nexus@One-North, Singapore 138543, Singapore; eng_sing_lee@nhgp.com.sg

**Keywords:** survey, disability, education, chronic conditions, health-related quality of life

## Abstract

Objective: The study aims to estimate the prevalence of disability and the association of disability with socio-demographic correlates and health outcomes among the Singapore population. Methods: Face-to-face interviews were conducted with a representative sample of Singapore’s population. Using the Washington Group’s questionnaire, disability was defined using both the ‘standard’ (a lot of difficulty or higher in one or more domains) and the ‘wider’ threshold (of ‘some’ or greater difficulty). Data on socio-demographic correlates, self-reported lifestyle, physical activity, chronic conditions, and health-related quality of life were also collected. Results: The prevalence of any disability using the standard threshold was 3.1% (95% CI: 2.4–4.1). When separated by disability type, mobility (1.8%) was the most prevalent, followed by vision (0.8%), cognition (0.5%), hearing (0.3%), and self-care (0.2). In the adjusted regression analysis, lower education and unemployed and economically inactive status (versus employed) were significantly associated with disability. Conclusions: Although this prevalence is lower than other countries, it is a significant finding in terms of actual numbers and impact at both the individual and the societal levels. Our findings also highlight the need to strengthen health services and preventive interventions targeting older adults and those who are physically inactive to reduce the burden of disability in these groups.

## 1. Introduction

The International Classification of Functioning, Disability and Health (ICF) defines disability as ‘impairments in body functions and structures, activity limitations, and participation restriction’. Disability is seen as the outcome of the interaction between a person (with a health condition) and that person’s contextual factors (environmental and personal factors) [1]. The United Nations Convention on the Rights of Persons with Disabilities (UNCRPD) defines persons with a disability as ‘those with long-term physical, mental, intellectual or sensory impairments. These impairments in interaction with various barriers may hinder their full and effective participation in society on an equal basis with others’ [2]. In a retrospective analysis from the World Health Survey (WHS) of non-institutionalised populations in 54 countries, the age and sex standardised disability prevalence among adults was estimated to be 14%. However, it varied significantly between countries ranging from less than 1% to 30% across countries [3].

The Behavioral Risk Factor Surveillance System (BRFSS) survey that collects state data about United States (U.S.) residents regarding their health and health behaviours, also assesses disability. The survey includes questions about six disability types—hearing, vision, cognition, mobility, self-care, and independent living. Respondents who answer ‘yes’ to any of the questions are identified as having a disability. According to the 2016 BRFSS survey, 25.7% of U.S. adults reported some form of disability. Disability in mobility (13.7%) and cognition (10.8%) were most frequently reported, while disability in self-care (3.7%) was least frequent. The study identified several socio-demographic correlates of disability. Women had a higher prevalence of any disability (i.e., at least one of the six disability types assessed in the survey) than men, as did those in older age groups compared with younger ones. Respondents with higher household income levels reported a lower prevalence of any disability [4].

Persons with disabilities face several challenges that restrict their effective participation on an equal basis with others in society. Access to basic services like transport, education, employment and built spaces are often inadequate, thus placing them at risk of losing out on development and subject to rising inequality [5]. At the same time, the prevalence of disability is increasing globally due to population aging and the increase in chronic health conditions [6], making disability a significant public health concern. Therefore, establishing the prevalence and correlates of disability is an essential first step for public health programs to address the needs of persons with disabilities.

Singapore is a highly developed Southeast Asian country with one of the highest per capita GDP in the world. Despite being a small country with limited natural resources, Singapore focuses heavily on developing its human capital and resources [7]. At the same time, Singapore has been striving to become an inclusive society where persons with disabilities (PWD) are respected, empowered, and enabled to achieve their full potential. The Enabling Masterplan Steering Committee (2004) defined PWD as “those whose prospects of securing, retaining places and advancing in education and training institutions, employment and recreation as equal members of the community are substantially reduced as a result of physical, sensory, intellectual and developmental impairments.” However, studies examining disability in Singapore’s population are few. A survey conducted by the National Council of Social Service among 2000 Singapore citizens and permanent residents aged 18 years and above in 2015 found that the self-reported disability prevalence rate was 3.4% among those aged 18–49 years and 13.3 among those aged 50 years and above [8]. Using data from the Retirement and Health Study (RHS), a longitudinal study of non-institutionalised older adults that used activities of daily living disability, Ng et al. [9] established the prevalence of mild, moderate, and severe disability among those aged 65 years and above as 9.6%, 6.6%, and 3.3% respectively. Another study conducted by Raghunathan et al. [10] examined the unmet needs of 100 people with disabilities. The authors found that informal care networks were the primary provider of caregiving support. Moreover, access to schooling was challenging, and PWD were subjected to bullying or isolation, received low incomes, and faced financial difficulties. They concluded that environmental and social factors played an important role in the disabling experiences faced by the participants. 

While these studies have reported the prevalence of disability using different scales and ascertained some of the challenges PWD face, there remains a need to establish the prevalence using a validated tool consistent with the ICF framework that can enable comparisons across countries. In addition, there exists a knowledge gap in the understanding of disability distribution across socio-demographic profiles and comorbidities in Singapore. Such data is essential for creating awareness and early identification of these conditions, along with strengthening services across healthcare and social welfare systems. Lastly, valid population disability data are necessary for evaluating the outcomes of programs and policies for PWD. 

The aim of the current study was thus to estimate the prevalence of disability, both overall and by functional disability types, and the association of disability with socio-demographic correlates and health outcomes among the Singapore population using the Washington Disability Questionnaire [11,12].

## 2. Materials and Methods

This cross-sectional study utilised data from a nationwide study that aimed to examine the knowledge, attitudes, and practices (KAP) regarding diabetes in the general population of Singapore [13]. Concurrently, an assessment of disability was included to understand the overall prevalence of disability in the general population and among different subgroups of the population. The current article focuses on the prevalence of disability and its correlates.

The study population comprised Singapore citizens and permanent residents aged 18 years and above who could understand and speak in one of the official languages of Singapore—English, Chinese, Malay, or Tamil. The study excluded those who were uncontactable due to incomplete or incorrect addresses, those living outside of the country, participants who were assessed to have difficulties in understanding the questionnaire due to cognitive difficulties as well as those with severe physical or mental disorders who were unable to answer the questionnaire on their own. Trained lay interviewers conducted the survey, and data were collected via computer-assisted personal interviews (CAPI) using handheld tablets. Participants were recruited between February 2019 to September 2020. 

### 2.1. Sampling

Based on data from an earlier study that examined knowledge of diabetes and risk factors [14], a target sample size of 3000 was estimated to meet the primary aims of the study. A disproportionate stratified sampling design was adopted wherein the proportion of respondents in each ethnic group (Chinese, Malay and Indian) was set to approximately 30%, as well as those belonging to the older age group were oversampled. The sample was drawn from a national administrative database of all residents in Singapore, which is updated regularly. For the results to be representative of the Singapore population, all estimates were analysed using survey weights to adjust for age and ethnicity post-stratification, oversampling, and non-response. The protocol of the study methodology has been published in an earlier article [13]. 

Ethical approval for the study was obtained from the Institute of Mental Health’s Institutional Research Review Committee and the National Healthcare Group’s Domain Specific Review Board (NHG DSRB Ref 2018/00430). All participants gave written informed consent, and parental consent was sought for those aged 18 to 20 years as the official age of majority in Singapore is 21 years and above. 

### 2.2. Questionnaires

#### 2.2.1. Disability Assessment

The study used a set of five questions from the short set of six questions of the Washington Questionnaire. This questionnaire was developed by the Washington Group on Disability Statistics and is consistent with the conception and principles of the definition of disability as articulated by the ICF [11,12]. The questionnaire in the current study comprised the following questions (1) do you have difficulty seeing, even if wearing glasses? (2) do you have difficulty hearing, even if using a hearing aid? (3) do you have difficulty walking or climbing steps? (4) do you have difficulty remembering or concentrating? (5) do you have difficulty (with self-care such as) washing or dressing? The last item on the short set questionnaire—‘Using your usual (customary) language, do you have difficulty communicating, (for example, understanding or being under-stood by others)?’ was not included in this survey as the study needed written informed consent and participants needing proxy informants were excluded from the study. Thus, participants with communication difficulties were ineligible for the study. 

Each of these questions had four ordinal responses of ‘no difficulty, some difficulty, a lot of difficulty, or cannot do at all’. Disability was assessed based on the responses of the participants in two ways:Standard Threshold—those who reported ‘a lot of difficulty’ and ‘cannot do at all’ were considered as having a disability, while those reporting ‘no difficulty’ and ‘some difficulty’ were not regarded as having a disability, as recommended by the developers [12].Wider Threshold- those reporting ‘some difficulty’, ‘a lot of difficulty’, and ‘cannot do at all’ were considered as having a disability [15].

While the first definition ensures comparability of our results with other international studies, the second, less conservative definition prevents missing respondents who may under-report the extent of disability fearing stigma or discrimination [16].

#### 2.2.2. Chronic Conditions Checklist

Information about chronic physical conditions was obtained using a checklist that has been used previously in Singapore-based studies [17]. Based on their responses, participants were further categorised as those with (i) no chronic physical conditions, (ii) one chronic physical condition, and (iii) multimorbidity (i.e., two or more chronic physical conditions). 

#### 2.2.3. Short Form (SF)-12

The Short Form Health Survey version 2 (SF-12v2^®^, Quality Metric Incorporated, Johnston, RI, USA) is a 12-item self-report instrument that assesses health-related quality of life (HRQOL) [18]. The SF-12v2 covers eight sub-domains: general health (GH), physical functioning (PF), role physical (RP), bodily pain (BP), vitality (VT), social functioning (SF), role emotional (RE), and mental health (MH). These 12-items are summarised into two composite summary scores—a physical component summary score (PCS) and a mental component summary score (MCS) which reflect physical and emotional health-related QOL, respectively [19]. 

#### 2.2.4. Global Physical Activity Questionnaire 

The Global Physical Activity Questionnaire (GPAQ) is a 16-item instrument developed by the World Health Organization to measure physical activity [20]. The frequency and duration of time spent doing physical activity are measured in three domains: activity at work, travel to and from places, and recreational activities. The total energy expenditure measured in metabolic equivalents (MET)/minutes/week is computed by summing up all moderate- to vigorous-intensity physical activities performed at work, transport, and recreation. Based on the total MET/minutes/week, participants were classified into sufficiently active (MET ≥ 600) and insufficiently active (MET < 600) as defined by the World Health Organization [21].

#### 2.2.5. Lifestyle

Participants were asked to state how healthy their lifestyles were by choosing one of the four statements—(i) I have a very healthy lifestyle, (ii) I have a fairly healthy lifestyle, (iii) I think my lifestyle can be improved, (iv) I think my current lifestyle is not healthy. Those endorsing (i) or (ii) were classified as ‘having a healthy lifestyle’, and those agreeing with statements (iii) or (iv) were classified as ‘having an unhealthy lifestyle’.

A structured questionnaire was used to obtain socio-demographic information regarding age, gender, ethnicity, marital status, personal income, educational and employment status, and self-reported diagnosis of diabetes. 

#### 2.2.6. Statistical Analysis

The prevalence of any form of disability and the specific types of disabilities were expressed as weighted percentages with 95% confidence intervals (CI). In addition, socio-demographic characteristics of the population (age, sex, ethnicity, marital status, employment status, and personal monthly income) stratified based on disability were also reported. The weighted mean and standard error were calculated for continuous variables, while unweighted frequencies and weighted percentages were presented for categorical variables. 

The associations between disability and socio-demographic characteristics were examined using binary logistic regressions, where disability served as the dependent variable, and socio-demographic characteristics served as independent variables. We also investigated the relationship between disability, lifestyle, and physical activity by using binary logistic regressions, analysing lifestyle and physical activity as the dependent variables and disability as an independent variable. An analysis of multinomial logistic regression was also conducted using the number of chronic conditions as the dependent variable and disability as the independent variable. Linear regressions were used to examine the association between PCS and MCS with disabilities, with PCS and MCS as the dependent variables and disability as the independent variables. Odds ratios (OR) and 95% confidence intervals (CI) were presented for binary logistic regressions. The multinomial logistic regression yielded prevalence ratios (PR) and 95% CI. For linear regressions, the beta-coefficient and 95% CI were reported.

All regression analyses were adjusted for socio-demographic characteristics. Standard errors were calculated using the Taylor series linearization method to account for complex survey design. All statistical analyses were performed using Stata/SE 15.0 (College Station, TX, USA), with two-sided tests assuming a 5% significance level. Missing data were handled via listwise deletion.

## 3. Results

A total of 2895 participants were included in the analysis. Using the standard threshold, the prevalence of any disability was 3.1% (95% CI: 2.4–4.1). When separated by disability type, mobility (1.8%, 95% CI: 1.3–2.6) was the most prevalent, followed by vision (0.8%, 95% CI: 0.4–1.4) and cognition (0.5%, 95% CI: 0.3–1.0) (Table 1). Table 2 shows the distribution of prevalence of overall disability (standard threshold), stratified by socio-demographic characteristics, lifestyle, physical activity, and the number of chronic conditions. The prevalence of specific disabilities (standard threshold), stratified by socio-demographic characteristics, lifestyle, physical activity, and the number of chronic conditions, is shown in Appendix A.

The prevalence of any disability using the wider threshold was 31.0% (95% CI: 28.6–33.4). When separated by disability type, cognition (17.2%, 95% CI: 15.2–19.3) was the most prevalent, followed by mobility (15.6%, 95% CI: 13.9–17.5) and vision (8.9%, 95% CI: 7.5–10.6) (Table 1). Appendix A shows the distribution of prevalence of overall disability (wider threshold), stratified by socio-demographic characteristics, lifestyle, physical activity, and the number of chronic conditions.

### 3.1. Associations between Socio-Demographic Characteristics and Disability

Table 3 shows the adjusted regression model examining the associations between socio-demographic characteristics and disability defined by the standard threshold. Education and employment status were significantly associated with disability in the adjusted regression analysis. 

Table 4 shows the adjusted regression model examining the associations between socio-demographic characteristics and disability defined by the wider threshold. Age, education, and monthly personal income were significantly associated with disability in the adjusted regression analysis.

### 3.2. Associations between Lifestyle, Physical Activity, Number of Chronic Conditions, PCS and MCS

Disability, as defined by the standard threshold, was significantly associated with physical activity, PCS and MCS after controlling for socio-demographic variables (Table 5). The odds of being insufficiently active (MET < 600) was 3.5 times (95% CI: 1.8–6.6) higher for PWD as compared to individuals with no disability. In addition, compared to individuals with no disability, the PCS of PWD was lower by 8.2 units (95% CI: −11.4 to −5.0), while the MCS score was lower by four units (95% CI: −6.8 to −1.1). 

Those with a disability, as defined by the wider threshold, were more likely to report having an unhealthy lifestyle than those with no disability (OR: 1.8, 95% CI: 1.3–2.4). The odds of being insufficiently active (MET < 600) was 1.8 times (95% CI: 1.3–2.4) higher for PWD as compared to individuals with no disability. The prevalence of having two or more chronic conditions was higher for PWDs than individuals with no disability (PR: 2.0, 95% CI: 1.4–2.7). The PCS of PWDs was 4.8 units lower (95% CI: −5.6 to −4.0), whereas their MCS was 4.2 units lower (95% CI: −5.2 to −3.2) (Detailed analysis available on request).

## 4. Discussion

The study established the overall prevalence of disability as 3.1% in the adult Singapore population using the standard threshold recommended by the developers. The three most prevalent disability types were mobility, vision and cognition, while the least prevalent disability was self-care. 

Despite using the same disability criteria (at least one severe or extreme difficulty with bodily functions), the prevalence may vary across studies depending on the number of domains assessed and the population’s age structure. Mitra and Sambamoorthi [3] analysed data from the World Health Survey (WHS) (2002–2004) and established the prevalence of disability to be 14% across all adults. The survey assessed disability based on four domains: seeing, concentrating (functioning impairment), moving around, and self-care (activity limitations/participation restriction), and the prevalence ranged from 2.3% in Ireland to 30% in South Africa. Using data from seven cross-sectional national surveys, and assessing disability across six functional domains, Mactaggart et al. [15] estimated the prevalence of disability as ranging from 3.2% in Vanuatu to 14.1% in Turkey. At the same time, a study in Saudi Arabia found that the prevalence of disability was 3.3% using two of the six questions from the Washington Group (WG) on Disability Statistics [22]. While the prevalence figures in Singapore are well within the range reported in the studies by Mitra and Sambamoorthi [3] and Mactaggart et al. [15], it is relatively low. In part, this may be because the participants of this study were required to provide written informed consent, i.e., they would have to understand the requirements of the research and the risk-benefits of participation before making their decision, which requires cognitive capacity. They also had to be capable of replying to the questions as proxy informants were not allowed. This may have resulted in the exclusion of those who had cognitive impairments from participating in the study and an over-representation of those with cognitive impairments among the non-responders, i.e., those who refused to participate in the study. The low figures as established in this study could also possibly reflect the highly developed economy of Singapore. Poverty is associated with lack of access to healthcare, malnutrition, poor living conditions, and unsafe work environments, all of which can increase the likelihood of disability [23,24]. Using data from the World Health Survey to compare the prevalence of disability and socioeconomic inequalities in disability, Hosseinpoor et al. [25] found that low and lower-middle-income countries had a higher prevalence of age- and sex-standardised disability than upper-middle- and high-income countries.

Using a wider threshold resulted in a 10-fold increase in the overall disability to 31.5% in the Singapore population. This increase was similarly seen in Mactaggart et al. [15] study when the authors used the wider threshold. In addition, the prevalence of any disability in the current study using the wider threshold was higher than that reported in the United States, of 25.7% from the 2016 BRFSS study [4]. BRFSS assessed six disability types (hearing, vision, cognition, mobility, self-care, and independent living), and those responding “yes” to ‘any difficulty’ in the six domains were classified as having the disability. This threshold corresponds to the wider threshold on the Washington Disability questionnaire. 

The prevalence of disability varied significantly across several socio-demographic groups in the univariate analysis. As expected, disability, as assessed by the standard threshold, increased with increasing age, with 8.4% of those aged 65 years and above reporting a disability versus 1.4% in those aged 18–34 years. The increasing prevalence of disability by age can be explained by increasing frailty, decreasing muscle strength, and reduced cardiopulmonary fitness [26,27], even in the absence of any disease. On examining specific disabilities, those aged 65 years and above had a higher prevalence of all disabilities except cognition and self-care. In contrast, those aged 35–49 and 50–64 reported the highest prevalence of disability in cognition and self-care, respectively. Older adults who are community-dwelling may be assisted by their family members or paid domestic helpers in their activities of daily living. Thus, they may not perceive self-care to be a disability. However, it is difficult to explain why those aged 35–49 years reported a higher prevalence of cognitive disability than the older age group. Individuals in this age group may be holding fast-paced and demanding jobs while balancing family expectations and other domestic responsibilities. These high expectations and stress could have led them to perceive some cognitive impairments. Those who were separated/widowed/divorced had a higher prevalence of disability than those who were single or married. The higher prevalence was seen across all disability types except hearing, where those who were married/cohabiting had a marginally higher hearing disability. Ling and Perry [28] suggested that disability affects both marriage formation and marriage dissolution. However, data from the current study did not indicate that disability decreased marriage formation. It is possible that disability led to marriage dissolution, or the loss of a spouse may have also resulted in less social support and encouragement of a healthy lifestyle, leading to disability. Similar trends in prevalence were observed for disability using the wider threshold.

Educational and employment status were associated with disability, according to the regression analysis that used disability defined by the standard threshold. The likelihood of disability was higher for those with lower education status as compared to those with a degree and higher education. The United Nations Convention on the Rights of Persons with Disabilities specifies that countries must ensure that PWD have access to an ‘inclusive, quality and free primary and secondary education’. However, data suggests that children and adolescents with a disability are less likely to complete primary education than their non-disabled peers and that these disparities increase at higher levels of education [29]. The under-identification or delays in identifying students with disabilities may lead to a lack of early intervention and support. This coupled with absenteeism, and lack of accommodation in schools, are possible factors for the lower educational achievement in this group [30]. However, it is also possible that lower education increases the risk of acquiring a disability [31], or individuals with higher education may cope better with their disability [32]. The cross-sectional nature of the study prevents us from establishing any causal relationships. Those who were economically inactive and unemployed were also significantly associated with disability. Globally, policymakers struggle with the social exclusion of people with disabilities in the labour market [33], and it is well known that persons with disability are underemployed [34]. Studies suggest that employers’ attitudes, discrimination and limited workplace accommodation may be factors that discourage the recruitment of workers with disability in the workforce [35,36].

Disability, as determined by the standard threshold, was significantly associated with insufficient physical activity and PCS and MCS even after controlling for significant confounders. Studies suggest that both self-efficacy and physical activity are important determinants of HRQOL [37,38,39]. Those with a disability may not feel confident about their abilities, and their disability may also limit mobility, thus impairing their engagement in physical activity, contributing to a reduced HRQOL. The MCS could also be associated with anxiety, depression, poor social support, or stigma experienced by PWD [40], but the current study did not include these measures. 

On the other hand, disability, as determined by the wider threshold, was associated with age, lower education, and lower-income. In addition, those with a disability as assessed by the wider threshold were more likely to report having an unhealthy lifestyle, being insufficiently active, having two or more chronic conditions, and lower HRQOL. Hence, including the ‘some’ category in the definition of disability can identify individuals with lower functioning who would benefit from support. However, the 10-fold increase in prevalence does pose challenges in terms of resource allocation and planning. Mactaggart et al. [41] have suggested that using a threshold of ‘a lot’ may be too restrictive, while the threshold of ‘some’ may be too broad to determine the prevalence of significant functional impairment. The researchers suggest a two-phase approach wherein populations are first screened with the Washington Group Questions to measure the prevalence of disability (“a lot of difficulty” or “cannot do”), which can then be used for cross-country comparisons. In the second phase, a simple clinical screen should be administered to all participants who respond to having at least “some” difficulty to ensure appropriate referrals and maximise functioning. Future prevalence studies in Singapore should consider using a clinical screen on those reporting ‘some difficulty’ to identify a sub-population that would need services. This approach would ensure a better allocation of resources and provision of services for those with functional limitations. Considering this in Singapore, our findings suggest that using a self-reported method in isolation may be overly restrictive at the threshold of “a lot” and too broad at the level of “some” to determine disability. With a self-reported tool and additional clinical screens for all who report “some” difficulty, we can identify the majority of people who experience either a moderate or severe clinical impairment or participation restriction.

Our findings are subject to several limitations. Firstly, the study had a response rate of 66%; thus, there may be a non-response bias. In addition, given the perceived burden of the entire survey, those with severe disabilities might have chosen not to participate in the survey. Secondly, all participants had to provide written informed consent to participate in the study, thereby excluding those with severe cognitive disabilities. Thirdly, there is an element of stigma related to having a disability, and thus respondents may tend to under-report it. Lastly, the study did not include those living in institutional settings (prisons and long-term nursing home residents), which might exclude persons with disabilities since persons residing in these settings might be more likely to have a disability. Thus, our findings are likely to be a conservative estimate of the disability prevalence in Singapore. Regardless, this is one of the few studies in Singapore that established the prevalence of overall and specific types of disability in the population. In addition, the study is among the first to examine the association of physical activity, chronic conditions, and HRQOL with disability in a national sample.

## 5. Conclusions

The study established the overall prevalence of disability in Singapore to be 3.1%. Although this prevalence is lower than other countries, it is a significant finding in terms of actual numbers and impact at both the individual and the societal levels. The 3rd Enabling Masterplan in Singapore has identified four key thrust areas: improving the quality of life of persons with disabilities, supporting caregivers, building the community, and building an inclusive society [42]. This Masterplan also made several recommendations, such as improving access to quality education and scaling up efforts to hire and manage employees with disabilities. The study found correlations between disability and lower education and unemployment, highlighting the need for further resources and concerted efforts to bridge this gap. Our findings also highlight the need to strengthen health services and preventive interventions targeting older adults and those who are physically inactive to reduce the burden of disability in these groups. In conclusion, future studies must consider a clinical screening of those reporting ‘some disability’ and include proxy informants to provide information that would enable more precise estimates of disability. This can aid the development of better policies, allocation of resources (including greater access to assistive devices), and services for this vulnerable population.

## Figures and Tables

**Table 1 ijerph-18-13090-t001:** Prevalence of Disability.

	**Standard Threshold**		**Wide Threshold**	
	**Prevalence % (95% CI)**	** *n* **	**Prevalence % (95% CI)**	** *n* **
Overall disability	3.1 (2.4–4.1)	128	31.0 (28.6–33.4)	1002
**Type of Disability**	**% (95% CI)**	** *n* **	**% (95% CI)**	** *n* **
Seeing	0.8 (0.4–1.4)	22	8.9 (7.5–10.6)	283
Hearing	0.3 (0.1–0.8)	5	3.4 (2.6–4.5)	118
Mobility	1.8 (1.3–2.6)	93	15.6 (13.9–17.5)	563
Cognition	0.5 (0.3–1.0)	24	17.2 (15.2–19.3)	524
Self-care	0.2 (0.1–0.6)	11	1.5 (1.0–2.3)	55

*n* represents the sample observations.

**Table 2 ijerph-18-13090-t002:** Prevalence of overall disability (standard threshold) by socio-demographic groups, lifestyle, chronic conditions and physical activity.

	Overall Disability	
	No Disability *n* (%)	Disability *n* (%)
**Age groups (years)**		
18 to 34	810 (98.6)	13 (1.4)
35 to 49	708 (98.5)	11 (1.5)
50 to 64	737 (96.2)	37 (3.8)
65 and above	512 (91.6)	67 (8.4)
**Sex**		
Female	1394 (97.2)	80 (2.8)
Male	1373 (96.5)	48 (3.5)
**Ethnicity**		
Chinese	770 (97.0)	26 (3.0)
Malay	918 (95.4)	56 (4.6)
Indian	878 (97.2)	40 (2.8)
Others	201 (98.0)	6 (2.0)
**Education**		
Primary and below	562 (91.8)	75 (8.2)
Secondary	655 (96.2)	29 (3.8)
Pre-U/Junior College	122 (97.6)	4 (2.4)
Vocational Institute/ITE	261 (97.6)	6 (2.4)
Diploma	473 (98.2)	6 (1.8)
Degree, professional certification, and above	694 (99.8)	8 (0.2)
**Marital Status**		
Single	715 (98.0)	16 (2.0)
Married/Cohabiting	1787 (97.1)	73 (2.9)
Separated/Widowed/Divorced	264 (91.7)	39 (8.3)
**Employment status**		
Employed	1898 (98.4)	35 (1.6)
Economically Inactive	746 (93.6)	83 (6.4)
Unemployment	123 (91.0)	10 (9.0)
**Monthly income in SGD (Personal)**		
Below 2000	1145 (94.4)	91 (5.6)
2000 to 5999	1001 (98.6)	15 (1.4)
6000 and above	296 (99.0)	4 (1.0)
No income	203 (95.1)	16 (4.9)
**Lifestyle**		
Healthy Lifestyle	1311 (96.6)	59 (3.3)
Unhealthy Lifestyle	1454 (97.1)	69 (2.9)
**Chronic conditions**		
No chronic condition	1213 (98.4)	16 (1.6)
One chronic condition	738 (97.8)	28 (2.2)
At least two or more chronic conditions	810 (93.8)	82 (6.3)
**Physical Activity**		
Sufficiently active MET ≥ 600	2355 (97.8)	74 (2.2)
Insufficiently active MET < 6000	410 (92.4)	54 (7.6)

*n* represents the sample observations.ITE: Institute of Technical Education; MET: Metabolic Equivalents; SGD: Singapore Dollars. Missing values: Marital Status (*n* for “No disability” = 1), Personal Monthly Income (*n* for “No disability” = 122, *n* for “Disability” = 2), Lifestyle (*n* for “No disability” = 2), Chronic conditions (*n* for “No disability” = 6, *n* for “Disability = 2), Physical Activity (*n* for “No disability” = 2).

**Table 3 ijerph-18-13090-t003:** Logistic regression with overall disability (standard threshold) as outcome and socio-demographics as independent variables.

	Overall Disability (Unadjusted)			Overall Disability (Multivariable)		
	OR	95% CI	*p*-Value	OR	95% CI	*p*-Value
**Age groups (years)**						
18 to 34 (Reference)						
35 to 49	1.1	0.3–3.4	0.915	2.3	0.4–12.0	0.342
50 to 64	2.7	1.1–6.9	**0.032**	2.4	0.6–10.0	0.234
65 and above	6.5	2.7–15.8	**<0.001**	3.0	0.7–12.3	0.133
**Sex**						
Female (Reference)						
Male	1.2	0.7–2.2	0.465	1.9	1.0–3.7	0.064
**Ethnicity**						
Chinese (Reference)						
Malay	1.6	1.0–2.6	0.070	1.4	0.8–2.5	0.203
Indian	1.0	0.6–1.6	0.870	1.2	0.7–2.1	0.460
Others	0.7	0.3–1.8	0.442	1.5	0.4–5.4	0.493
**Education**						
Degree, professional certification, and above (Reference)						
Primary and below	37.8	15.6–91.2	**<0.001**	24.2	5.1–113.4	**<0.001**
Secondary	17.0	6.4–45.1	**<0.001**	15.0	3.1–71.5	**0.001**
Pre-U/Junior College	10.5	2.1–53.0	**0.005**	8.8	1.3–59.2	**0.025**
Vocational Institute/ITE	10.4	2.5–42.8	**0.001**	12.1	2.1–69.2	**0.005**
Diploma	7.9	2.2–28.4	**0.001**	9.9	1.7–58.8	**0.012**
**Marital Status**						
Married/Cohabiting (Reference)						
Single	0.7	0.3–1.5	0.351	1.5	0.4–5.4	0.534
Separated/Widowed/Divorced	3.0	1.5–6.2	**<0.001**	1.9	0.9–4.1	0.112
**Employment status**						
Employed (Reference)						
Economically Inactive	4.2	2.2–7.9	**<0.001**	2.9	1.3–6.6	**0.011**
Unemployment	6.1	2.1–17.5	**0.001**	4.8	1.2–18.9	**0.024**
**Monthly income SGD (Personal)**						
Below 2000 (Reference)						
2000 to 5999	0.2	0.1–0.5	**0.001**	0.7	0.3–2.1	0.57
6000 and above	0.2	0.0–0.8	**0.028**	1.7	0.3–10.4	0.57
No income	0.9	0.3–2.3	0.797	0.7	0.2–2.2	0.54

ITE: Institute of Technical Education; SGD: Singapore Dollars. *p*-value < 0.05 are in bold.

**Table 4 ijerph-18-13090-t004:** Logistic regression with overall disability (wide threshold) as outcome and socio-demographics as independent variables.

	Overall Disability			Overall Disability		
	(Unadjusted)			(Multivariable)		
	OR	95% CI	*p*-Value	OR	95% CI	*p*-Value
**Age groups (years)**						
18 to 34 (Reference)						
35 to 49	1.4	1.0–2.0	**0.042**	1.7	1.1–2.6	**0.022**
50 to 64	2.2	1.6–3.0	**<0.001**	1.8	1.1–3.0	**0.012**
65 and above	4.8	3.3–6.8	**<0.001**	3.1	1.8–5.4	**<0.001**
**Sex**						
Female (Reference)						
Male	0.8	0.7–1.1	0.132	0.9	0.7–1.2	0.703
**Ethnicity**						
Chinese (Reference)						
Malay	1.3	1.1–1.6	**0.007**	1.3	1.0–1.6	0.066
Indian	0.9	0.7–1.1	0.220	1.0	0.8–1.2	0.858
Others	0.8	0.6–1.2	0.368	1.2	0.8–1.8	0.420
**Education**						
Degree, professional certification, and above (Reference)						
Primary and below	3.9	2.8–5.6	**<0.001**	1.6	1.0–2.6	0.058
Secondary	2.7	1.9–3.9	**<0.001**	1.7	1.1–2.6	**0.016**
Pre-U/Junior College	1.2	0.6–2.3	0.598	0.8	0.4–1.6	0.623
Vocational Institute/ITE	1.9	1.2–3.1	**0.008**	1.7	1.0–2.9	0.062
Diploma	1.7	1.1–2.4	**0.010**	1.5	1.0– 2.3	0.053
**Marital Status**						
Married/Cohabiting (Reference)						
Single	0.6	0.5–0.8	**0.001**	1.0	0.6– 1.4	0.817
Separated/Widowed/Divorced	2.0	1.4–3.0	**<0.001**	1.4	1.0–2.2	0.073
**Employment status**						
Employed (Reference)						
Economically Inactive	2.2	1.7–2.8	**<0.001**	1.4	1.0–2.0	0.058
Unemployment	1.9	1.1–3.3	**0.018**	1.7	0.9–3.0	0.084
**Monthly income SGD (Personal)**						
Below 2000 (Reference)						
2000 to 5999	0.4	0.3–0.6	**<0.001**	0.69	0.49– 0.96	**0.028**
6000 and above	0.4	0.2–0.5	**<0.001**	0.7	0.4–1.2	0.174
No income	0.8	0.5–1.3	0.462	0.8	0.5–1.3	0.375

ITE: Institute of Technical Education; SGD: Singapore Dollars. *p*-value < 0.05 are in bold.

**Table 5 ijerph-18-13090-t005:** The association of overall disability (standard threshold) with lifestyle, physical activity, chronic conditions, PCS and MCS.

	Unhealthy Lifestyle	Insufficiently Active (MET)	Chronic Conditions 1 ≥ 2		PCS	MCS
	Adjusted *					
	OR (95% CI)		PR (95% CI)		B-coeff (95% CI)	
No disability (ref)						
Disability	1.2 (0.7–2.3)	3.5 (1.8–6.6)	1.2 (0.5–3.1)	2.0 (0.9–4.6)	−8.2 (−11.4–−5.0)	−4.0 (−6.8–−1.1)

* Adjusted for age group, sex, ethnicity, education, marital status, employment status and monthly income (personal). MET: Metabolic Equivalents; MCS: Mental Component Summary Score; OR: Odds Ratio; PCS: Physical Component Summary Score; PR: Prevalence Ratio.

## Data Availability

The datasets generated during and/or analysed during the current study are available from the corresponding author on reasonable request.

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
