# Peer review of "The Prevalence and Correlates of Disability in Singapore: Results from a Nationwide Cross-Sectional Survey"

_ijerph, 2021, doi:10.3390/ijerph182413090_

Round 1

Reviewer 1 Report

This study is well constructed and the article is very well written. While it doesn't tell us much that we didn't expect - the main findings are that disability is associated with things like lack of physical activity, unemployment, low education, and low income - it makes a good contribution to the literature given the paucity of studies on disability in Singapore. There is also a useful section of discussion about the Washington Group’s questionnaire which will have an impact on scholars using that methodology and other ones that are similar. One could offer a criticism that this seems to be just a sliver of a much bigger study, but the data included here is appropriate and the discussion more than fulsome enough to warrant its own  article.

            I really don't have a lot of comments to make, which itself speaks to the quality of the submission (although I'm not a quantitative scholar, so can't comment upon the use of statistics and their appropriateness).

I would suggest a little more discussion of the exclusion criteria would be good, given the low prevalence of cognitive disabilities (0.5%). There are a couple of lines in the limitations paragraph at the end, but I think it's worth discussing this in the main discussion section. I was struck that the criteria were likely to severely limit the number of people with cognitive disabilities who could participate in this study, and think that's worth a little more discussion.

            One question that I have, which would likely be more a suggestion for future studies, or something to include at the end, is some consideration that Singapore is (as mentioned p.2, lines 67-70), a very wealthy society. If one looks at the literature on safety and accidents and injury (which lead not to just higher mortality, but also higher rates of disability), we know that societies that are overall less economically well off, or that have a very high wealth disparity, tend to have higher rates of accidents.  Poorer people tend to work in more dangerous jobs, live in less safe environments, etc. so this may be part of the reason why we see the fairly low prevalence of disability that shows in this study. I leave it to the authors to make what they want of this suggestion.

Author Response

We would like to thank the reviewers for their extremely encouraging and insightful comments. The comments have helped us further improved our article.  Our response is as follows:

Reviewers' comments

This study is well constructed and the article is very well written. While it doesn't tell us much that we didn't expect - the main findings are that disability is associated with things like lack of physical activity, unemployment, low education, and low income - it makes a good contribution to the literature given the paucity of studies on disability in Singapore. There is also a useful section of discussion about the Washington Group’s questionnaire which will have an impact on scholars using that methodology and other ones that are similar. One could offer a criticism that this seems to be just a sliver of a much bigger study, but the data included here is appropriate and the discussion more than fulsome enough to warrant its own  article.

Thank you so much for your encouraging comments. We wanted to keep this article focussed and we decided to pursue this line of analysis, but if it is not against the journal’s policies, we would like to hear from you about your thoughts on a larger article. While we agree it would be a separate article, we feel it could be important.

I really don't have a lot of comments to make, which itself speaks to the quality of the submission (although I'm not a quantitative scholar, so can't comment upon the use of statistics and their appropriateness).

I would suggest a little more discussion of the exclusion criteria would be good, given the low prevalence of cognitive disabilities (0.5%). There are a couple of lines in the limitations paragraph at the end, but I think it's worth discussing this in the main discussion section. I was struck that the criteria were likely to severely limit the number of people with cognitive disabilities who could participate in this study, and think that's worth a little more discussion.

We agree with the reviewer’s comments. Since this was part of a larger study where consent and understanding of the questionnaires was a pre-requisite; this would have limited the participation of people with severe cognitive disability. We have incorporated this possibility in the revised manuscript.

            One question that I have, which would likely be more a suggestion for future studies, or something to include at the end, is some consideration that Singapore is (as mentioned p.2, lines 67-70), a very wealthy society. If one looks at the literature on safety and accidents and injury (which lead not to just higher mortality, but also higher rates of disability), we know that societies that are overall less economically well off, or that have a very high wealth disparity, tend to have higher rates of accidents.  Poorer people tend to work in more dangerous jobs, live in less safe environments, etc. so this may be part of the reason why we see the fairly low prevalence of disability that shows in this study. I leave it to the authors to make what they want of this suggestion.

This is an important point that is raised by the reviewer, and we have added it to our discussion.

Reviewer 2 Report

This is a cross-sectional study with impact on Singapore healthcare system. From the introduction, it seems that the reason for doing this study is not clear. Lack of such study in Singapore is not strong enough to justify the reason of doing this. It is also better to include the impact of knowing the disability situation in Singapore in the introduction part.

It is quite confused that whether this study is using the data from a diabetes group or from general population. Are the data only included diabetes patients? 

What program recruited the participants between 2019 and 2020?

The data analysis method was valid.

May the author explain why healthy and unhealthy lifestyle is nearly half-half, but sufficient active persons are so much higher than insufficient active persons? What did this indicate?

The explanation of findings is well presented in discussion part. However, it is still unclear about the implication of this study. Even the proportion of disability is low, anything we can do to further prevent the presence of disability? When disability occurs, as suggested by the associated variables, what should we do to help these disability persons?

Author Response

We would like to thank the reviewers for their extremely encouraging and insightful comments. The comments have helped us further improved our article.  Our response is as follows:

This is a cross-sectional study with impact on Singapore healthcare system. From the introduction, it seems that the reason for doing this study is not clear. Lack of such study in Singapore is not strong enough to justify the reason of doing this. It is also better to include the impact of knowing the disability situation in Singapore in the introduction part.

We have revised our introduction to address the reviewer’s comments.

It is quite confused that whether this study is using the data from a diabetes group or from general population. Are the data only included diabetes patients? 

What program recruited the participants between 2019 and 2020?

We apologise for the confusion. Our Methods section clearly states that the study is a population-based study and not a study on patients with diabetes. The main intent of the study was to determine the knowledge attitudes and practices of the Singapore population towards diabetes. We have revised it further to make this clearer.

The data analysis method was valid.

May the author explain why healthy and unhealthy lifestyle is nearly half-half, but sufficient active persons are so much higher than insufficient active persons? What did this indicate?

This is an interesting observation which we will be explicating in a separate paper as we have decided that it would be beyond the scope of this present paper. In reply to this reviewer’s query, we would like to say that healthy lifestyle is a broad and multi-dimensional concept that includes diet, stress levels, sleep, and work-life balance – other than physical activity.

The explanation of findings is well presented in discussion part. However, it is still unclear about the implication of this study. Even the proportion of disability is low, anything we can do to further prevent the presence of disability? When disability occurs, as suggested by the associated variables, what should we do to help these disability persons?

We thank the reviewer for this suggestion and have addressed the comments in the revised discussion.

Round 2

Reviewer 2 Report

Thank you for the revised manuscript and your responses to the reviewers' comments. After re-reading the manuscript and checking for the changes, I find that many of them are satisfactorily addressed. However, there are some minor mistakes.

Please proofread the revised part, such as from line 295 to 311, "that was reported by in the studies by", "low and lower middle-income countries".

Author Response

Thank you for the revised manuscript and your responses to the reviewers' comments. After re-reading the manuscript and checking for the changes, I find that many of them are satisfactorily addressed. However, there are some minor mistakes.

Please proofread the revised part, such as from line 295 to 311, "that was reported by in the studies by", "low and lower middle-income countries".

We apologise for the error and have proofread the revised part.